# Using Artificial French Data to Understand the Emergence of Gender Bias in Transformer Language Models

**Lina Conti**🦕,🦩,🐣 and **Guillaume Wisniewski**🐣

🦕 Fondazione Bruno Kessler, Italy
🦩 University of Trento, Italy
🐣 Université Paris Cité, LLF, CNRS, 75 013 Paris, France
lvarellaconti@fbk.eu, guillaume.wisniewski@u-paris.fr

## Abstract

Numerous studies have demonstrated the ability of neural language models to learn various linguistic properties without direct supervision. This work takes an initial step towards exploring the less researched topic of how neural models discover linguistic properties of words, such as gender, as well as the rules governing their usage. We propose to use an artificial corpus generated by a PCFG based on French to precisely control the gender distribution in the training data and determine under which conditions a model correctly captures gender information or, on the contrary, appears gender-biased.

## 1 Introduction

Numerous studies have demonstrated NLMs' ability to capture linguistic properties, such as grammatical categories and syntactic rules like number and gender agreement, without any direct supervision (Belinkov et al., 2020; Manning et al., 2020). However, they leave unanswered the fundamental question of how NLMs uncover these properties from raw textual data alone. This study aims to address this question by focusing on the specific case of gender in French.

Our motivation for working on gender is twofold. Firstly, the presence of gender bias in NLMs raises significant societal concerns (Gaucher et al., 2011; Cao and Daumé III, 2020) and understanding how these models capture and use gender information is a crucial step towards addressing such biases. Secondly, the grammatical expression of gender in languages such as French makes it possible to construct testbeds to explore how this linguistic information can emerge in an unsupervised way in LMs.

This work addresses two questions: *i)* whether the gender that should be assigned to a word is ingrained in its representation or dynamically computed based on the linguistic context and *ii)* which characteristics of the training set, such as word frequency or gender imbalance, affect an NLM's ability to capture gender information. By answering these two questions, we hope to enhance our comprehension of how an NLM determines a word's gender (or other linguistic properties) and gain insights into the conditions that give rise to a biased representation of gender in these models.

To achieve this, we propose to use artificial corpora generated by PCFGs (§2).[1] This approach allows us to have complete control over the gender of nouns and the contexts in which they are presented. Following the methodology from Kim and Linzen (2020), we evaluate the NLM's capacities on a test set specifically designed to contain examples that differ from those of the training set based on well-defined criteria. By precisely manipulating the contexts in which words appear when training and testing, we can establish causal relations between changes in these contexts and the NLM's ability to assign gender.

While the scope of this study is limited to examining the impact of certain training data characteristics on NLMs' gender learning, our aspiration is that it serves as an important step in a larger process towards understanding and rectifying biased behaviour in NLMs. The subsequent phase entails evaluating how alterations in the training corpus and therefore on how gender is learnt manifest in the text generated by these models. Once a causal relationship is established between potential factors influencing gender learning and the behaviour of NLMs, the objective is to use this knowledge to mitigate bias in "real" language models trained on natural data by specifically targeting the factors identified as playing an important role in our controlled experiments.

Contrary to prior research, our results (§3) show that transformer language models do not necessarily amplify gender imbalances present in the data. Moreover, we observe the presence of gender bias even in cases where the training data is perfectly balanced.

---

[1]Code available at: https://github.com/lina-conti/artificial-data-gender.

## 2 Observing the Emergence of Gender Information in NLMs

To investigate how NLMs discover gender information, we propose to train models on controlled sets where gender expression is precisely regulated. We then use a linguistic probe (Belinkov, 2022), to assess the model's ability to learn this property in various scenarios. Following a current line of research (Kim and Linzen, 2020; White and Cotterell, 2021), we use artificial language corpora generated by PCFGs to have full control over the expression of gender. This control encompasses, for example, the frequency of each gender and the informativeness of word occurrences' context with regard to gender.

Our PCFGs aim to mimic a subset of the French language focusing on the expression of gender within noun phrases (NPs). While some prior studies in the literature (Kim and Linzen, 2020) aim to maintain semantic coherence in their artificial data, others (Gulordava et al., 2018) leverage the absence of such coherence to disentangle syntax from semantics. In this work, we chose to focus solely on the grammatical aspect of gender to eliminate potential confounding factors introduced by semantics, such as stereotypes propagated by the distributional environment of words. This deliberate decision narrows down the potential sources of bias under investigation, at least for the present. When constructing the syntax of our language, we also prioritise simplicity over realism, particularly concerning elements unrelated to the specific aspects we investigate in relation to gender learning.

Our main motivation for selecting French as the basis for our PCFGs stems from the opportunity it provides to construct contexts that either mark gender or not. French has two grammatical genders: feminine and masculine.[2] Nouns, determiners, adjectives and pronouns can have fixed genders or be epicene, meaning they can denote individuals of any gender. In this case, the gender must be inferred from the context. For instance, when the epicene noun "artiste" is accompanied by the feminine determiner "une", a feminine gender can be inferred. However, when used with an epicene determiner as in "chaque artiste," the gender becomes ambiguous since the context provides no information about it.

The noun phrase generation in the PCFGs involves three levels of rules. The first level determines whether the noun occurs in a context where gender information is revealed by determiners and adjectives or in a gender-ambiguous context:[3]

$$\text{NP} \rightarrow \text{NP}_{\text{gendered}} \mid \text{NP}_{\text{ambiguous}} \qquad (1)$$

The second level of rules, following a similar structure, describes the constituents of each of these contexts. Meanwhile, the third level establishes the lexical rules that associate terminal symbols (words) to respective word categories. For example:

$$\text{NP}_{\text{ambiguous}} \rightarrow \text{DET}_{\text{epic}} \left( \text{ADJ}_{\text{epic}} \mid \epsilon \right) \text{NOUN} \qquad (2)$$
$$\text{DET}_{\text{epic}} \rightarrow \texttt{chaque} \mid \texttt{leur} \mid \ldots \qquad (3)$$

Through the manipulation of rules and their probabilities, we can determine the gender of nouns and exert precise control over the gender-related information accessible to the model during training (e.g., by ensuring that some nouns only appear in gender-neutral contexts). We can then use the following experimental pipeline to assess the LM's capacity to assign a gender to nouns in different scenarios:

1. we generate two train sets (`lm_train` and `probe_train`) and a test set (`probe_test`) using slightly different PCFGs;
2. we train a LM on `lm_train`;
3. using `probe_train`, we train a linguistic probe (Belinkov, 2022) to predict the gender (masculine or feminine) to be assigned to noun occurrences based on their contextualised representation at the final layer of the LM;
4. we evaluate the probe's performance on `probe_test`.

Hewitt and Liang (2019) highlight that by relying solely on probe accuracy we cannot tell whether a high score means the information is encoded in the representations or whether the probe has just learnt the task. In order to avoid this problem and ensure that a high probe accuracy indicates the encoding of gender information within the language model's representations, nouns present in `probe_train` are excluded from `probe_test`.

In all our experiments, we used the transformer LM from White and Cotterell (2021) (see §B). We generated $10,000$ sentences for `lm_train`, $1,000$ for `probe_train` and $1,000 \times n$ for `probe_test` where $n$ represents the number of distinct groups considered in our analyses.

All aspects of French syntax relevant to establishing our PCFGs, and those regarding gender

---

[2]Proposals exist for gender-neutral morphosyntactic alternatives in French, such as the neo-pronoun *iel*. However, since as of now their adoption remains limited, we align our PCFGs with standard French, which recognises only two genders.

[3]For clarity, we omit rule probabilities.

in particular, are very similar to other romance languages, such as Spanish, Portuguese and Italian. Changing the lexical rules would be enough to make the PCFGs generate sentences emulating another one of those languages. Since such a PCFG would be equivalent from the point of view of the NLM, the results of our experiments can be generalised to these other languages. However, different PCFGs would be necessary to study how NLMs learn gender in languages with vastly different gender systems, such as languages with more than two genders or languages that have no gender system (Corbett, 2013). The exploration of this topic is deferred to future research.

## 3   Experimental Results

**Decoupling contextual gender information from static gender associations**   Our first experiment aims to assess whether NLMs can use contextual information during inference to find the gender of words that occur only in ambiguous contexts during training, thus assessing whether the model's representation of gender relies on information memorised from training data or is dynamically constructed during inference based on linguistic cues.

| Training context | Inference context | |
|---|---|---|
| | Ambiguous | Gendered |
| Ambiguous | 50.37 ±0.49 | 95.84 ±0.20 |
| Gendered | 99.26 ±0.08 | 99.99 ±0.01 |

Table 1: Accuracy (in %) when predicting the gender of words according to whether they occurred in gendered or ambiguous contexts in `lm_train` and `probe_test`.

More precisely, we used a PCFG (Appendix A.1) where all nouns are either masculine or feminine and consistently appear in either gendered or non-gendered contexts within a given dataset. Table 1 reveals that gender identification is nearly flawless for words encountered in gendered contexts in `lm_train`. Even when these words appear in ambiguous contexts in `probe_test`, the model's identification of gender remains highly accurate. This suggests that gender information is inherently encoded within the model's representations, reflecting the associations it has learned during the training process.

The model exhibits a reasonable ability to infer gender solely from context for words whose gender was not revealed in `lm_train`. This highlights the model's capacity to extract gender information from the sentence structure. As expected, for words

that exclusively appear in ambiguous contexts, the model's performance is closer to chance level.

Our observations reveal a potential cause of bias emergence: when a word is consistently associated with a specific gender in the training set, that gender becomes ingrained in its representation. It may then be challenging to revise this association when contextual cues contradict the learnt gender. This leads to language errors when a model relies on stereotypes rather than context, for example, assigning "secretary" a female gender even when the syntax of the sentence indicates that the word refers to a masculine entity (Zhao et al., 2018). The presence of this form of gender bias in natural language models that are deployed in industry leads to representational harms (Crawford, 2017), by perpetuating stereotypes associated to each gender and contributing to confine individuals in restrictive gender roles.

**Assignment of gender to epicene nouns**   To gain deeper insights into the relationship between gender encountered during training and gender indicated in the context, we incorporated epicene nouns into our PCFG. This allowed us to investigate the impact of conflicting genders between the model's training data and the context during inference.

More specifically, we constructed a PCFG (Appendix A.2) that includes five noun categories: feminine nouns, masculine nouns and three categories of epicene nouns. Each of the five noun categories appears in both gendered and ambiguous contexts in `lm_train`. When they appear in gendered contexts, one epicene category has an equal likelihood of being associated with either gender; another has a 25% chance of appearing in a feminine context and a 75% chance of appearing in a masculine one; the third category has the opposite probabilities. We evaluated the NLM's ability to identify the gender of the referent of nouns in these five categories using the same methodology as in the previous experiment.

As reported in Figure 1, the model demonstrates remarkable accuracy in determining the gender of the noun's referent based on contextual cues when it appears in a gendered context. It consistently assigns a feminine gender when the noun appears in a feminine context, achieving an accuracy rate close to 100%, and hardly ever assigns a feminine gender when the noun is in a masculine context, maintaining an assignment rate close to 0%. This holds true for all categories, even when the noun consistently appeared in the opposite gender context during training. However, in ambiguous

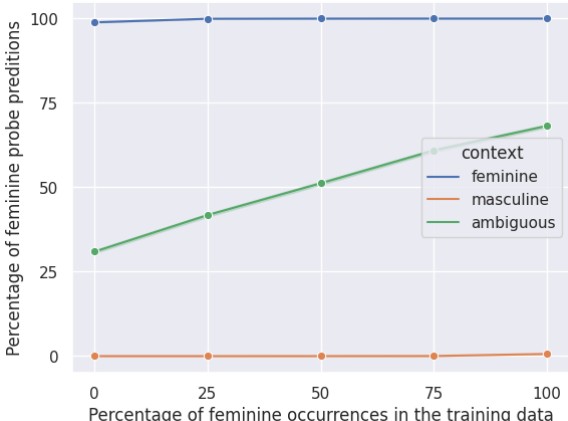

Figure 1: Percentage of feminine probe predictions for epicene nouns by percentage of feminine occurrences of those nouns in `lm_train` and context in which they appear in `probe_test`.

| Noun frequency | Inference context | |
| --- | --- | --- |
| | Ambiguous | Gendered |
| very rare | 73.87 ±0.48 | 98.40 ± 0.14 |
| rare | 74.80 ±0.45 | 98.08 ± 0.14 |
| frequent | 76.25 ±0.42 | 98.13 ± 0.13 |
| very frequent | 77.30 ±0.36 | 98.19 ± 0.11 |

Table 2: Probe accuracy (%) by noun's frequency in `lm_train` and context in `probe_test`.

contexts, the model tends to assign the noun its most frequently associated gender in `lm_train`.

Unlike the observations by Stanczak and Augenstein (2021) for models trained on natural language, our model does not appear to amplify imbalances present in the training data. For instance, when epicene words were encountered in a feminine context 75% of the time during training, the model assigns a feminine gender to their referent only 60.93% ±0.48 of the time in an ambiguous context, rather than maintaining a 75% or higher assignment rate.

This observation indicates that the inability of LMs to assign alternative genders to epicene words, such as occupational nouns, which have been extensively studied to uncover gender bias in NLMs, cannot be solely attributed to a gender frequency imbalance in their occurrences in the training data.

**Impact of word frequency** We now explore the influence of word frequency on the ability of NLMs to capture gender information and examine whether models are more prone to making errors on words that occur less frequently in the training data. To investigate this, we categorised the nouns in `probe_test` based on their probability of being generated in `lm_train`: the 25% of nouns with the lowest probability of being generated were classified as "very rare", the subsequent 25% of nouns as "rare", and so on.

Table 2 presents the average probe accuracy for nouns in each quartile based on whether they appeared in a gendered or ambiguous context in the test set. The performance of the probe in ambiguous contexts exhibits a positive correlation with word frequency, indicating that assigning gender to words that have been encountered more frequently in the training data is relatively easier.

The difference in accuracy between very infrequent and very frequent nouns may not appear substantial, with a mean accuracy of 73.9% for very rare nouns compared to 77.3% for very frequent nouns. However, the absence of overlapping confidence intervals among the quartiles indicates that the influence of noun frequency on the occurrence of gender assignment errors is statistically significant.[4]

Accuracies for gendered contexts are more uniform across all quartiles, suggesting that word frequency plays a comparatively minor role. This observation suggests that when gender information is readily available in the context during inference, the impact of frequencies in the training corpus becomes less critical.

**Influence of the gender distribution on default gender guessing** We now turn to the impact of the imbalance in gender distribution in the training data, which is often cited as the reason for the overuse of masculine gender by NLMs. The disproportionally low representation of the feminine gender constitutes a form of bias referred to as under-representation, which results in the marginalisation and invisibility of women (Crawford, 2017). To assess the extent to which this over-assignment of masculine gender can be attributed to gender frequency imbalances in the training corpus, we conducted an experiment using five different PCFGs. These PCFGs assigned varying probabilities to feminine and masculine noun phrases, allowing us to generate training sets that ranged from perfectly gender-balanced to exhibiting a strong bias towards one gender. We then assessed the frequency with which models trained on data generated by each PCFG assigned each gender to nouns

---

[4]Given that our work is based on artificial data, we cannot make conclusive assertions regarding the magnitude of this effect for models trained on natural data. Nevertheless, employing artificial data enables us to identify the factors that can play a substantial role in gender assignment.

for which they had no explicit gender information.[5]

In an unbiased model, we would expect the probe to predict approximately 50% of nouns to be of each gender. However, if models exhibited a bias towards the majority class the probe would more frequently predict the gender that was more prevalent in the train set. We quantify the *degree of bias* as follows:

$$\frac{2 \times \text{number of majority class predictions}}{\text{total number of predictions}} - 1$$

A degree of bias of 0 indicates that the probe predicted 50% of nouns for each gender, while a value of 1 (resp. -1) suggests a strong bias toward the majority (resp. minority) class.

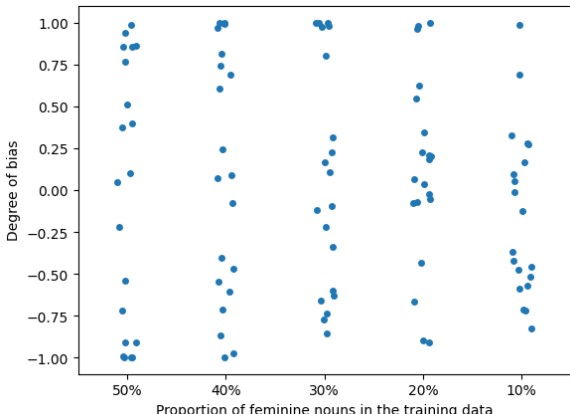

Figure 2: Degree of bias exhibited by models trained on corpora with different gender distributions.

Figure 2 displays a scatter plot illustrating the degree of bias observed in models trained on different corpora. Each corpus contained varying proportions of feminine noun phrases, ranging from 50% down to 10%, with the remaining noun phrases being masculine. For each gender distribution, twenty models were trained.

The degree of bias observed in models can vary significantly even within a single gender distribution. Models may exhibit bias towards the majority class, the minority class, or any point in between. Surprisingly, the gender distribution in the models' training data does not seem to have a substantial impact on the degree of bias. One possibility to explore in future research is whether the examples seen at the beginning of training have a disproportionate influence on the likelihood that the model attributes to each gender. These findings suggest that the origin of bias in language models is more complex than

initially thought. It raises the question of whether training models on balanced data, which has been explored as a strategy to mitigate gender bias, is sufficient, as there are likely other factors at play.

## 4 Conclusions

In this study, we use artificial language corpora to gain deeper insights into how NLMs acquire linguistic knowledge from unannotated texts. Our experiments shed light on the complex factors influencing the emergence of gender (as an illustrative linguistic property) in NLMs. It is crucial to underscore that, being derived from artificial data, our observations offer insights into the potential mechanisms involved in learning gender, but they do not provide definitive proof that these mechanisms are the core factors driving NLMs' ability to acquire and use linguistic knowledge when working with natural language data. However, our findings on NLMs' learning process can suggest future directions for bias mitigation. For instance, our results challenge the assumption that training data imbalance is the sole cause of gender bias in NLMs and thus shed doubt on the idea that training or fine-tuning on gender-balanced corpora would be enough to avoid gender bias. They also show that an imbalanced gender distribution in the training data does not necessarily lead a transformer-based LM to over-assign the majority class gender. This is cause for optimism as it suggests that transformers' learning process is not inherently biased or irremediably predisposed to reproducing the most frequent patterns. Our future work will concentrate on generalising our observations to models trained on natural language data.

## Acknowledgments

We sincerely thank the reviewers for their careful reviews and insightful comments, which were of great help in improving the manuscript. We would also like to gratefully acknowledge support from the Labex EFL (ANR-10-LABX-0083).

## 5 Limitations

The first limitation of our work concerns the use of probes. While we have carefully distinguished between information learnt from the probe's training process and information inherent to the model's representations as advocated by Hewitt and Liang (2019), it is worth noting that using probes introduces an additional system to examine gender, rather than directly studying the behaviour of LMs.

---

[5]These nouns exclusively appeared in gender-ambiguous contexts in both `lm_train` and `probe_test` and were not included in `probe_train`.

In future research, it would be valuable to explore more intricate PCFGs that could support the coreference resolution challenges usually used to highlight gender bias in NLMs, allowing us to compare gender assignment measurements obtained through this method with the results obtained using probes.

In addition to our previous considerations, the extent to which our findings can be extrapolated to "real" models trained on natural language also warrants further investigation. Current state-of-the-art LMs are significantly larger than the models used in our study and exhibit emergent capacities resulting from their increased size and training data (Manning et al., 2020; Wei et al., 2022). Therefore, it would be relevant to examine how model and dataset sizes impact our experiments. Additionally, comparing the variability of gender expression and the gender distributions in our data to those observed in natural language would offer valuable insights. By further exploring these points, we could gain a better understanding of the robustness and applicability of our findings to real-world language models.

It should also be kept in mind that while all the results described in this work are based on the analysis of an artificial corpus of data, the design of our PCFGs draws inspiration from the expression of gender in French. The expression of gender varies significantly across languages and exploring the ability of systems to learn more complex gender systems would be an interesting avenue for future research.

A final limitation of this work is the consideration of a binary gender system, which will be revisited in Section 6.

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

# 6 Ethical Issues

One ethical issue central to our work concerns the treatment of gender as a binary variable. While this aligns with the grammatical conventions of standard French, it falls short from a social perspective as it disregards individuals who do not identify within this binary framework. By imposing a binary choice, assumptions about gender need to be made, which can potentially result in inaccuracies, or the strategy of masculine default is adopted, further marginalising genders other than the masculine. Proposed solutions exist to make French more gender-inclusive, and it is crucial to contemplate their integration into NLP systems. Using artificial data, as we have done, provides an opportunity to explore the integration of gender-inclusive language in NLP, even in the absence of natural data, which is currently an obstacle. Therefore, artificial languages can serve as an initial step in determining the requirements for gender-inclusive language models, conducting experiments, for instance, to ascertain the number of gender-neutral language instances necessary for the model to comprehend non-binary gender.

# A PCFGs to model gender in French

Below, we present the complete set of non-terminal rules used in our experiments. To construct the terminal symbols, we draw from a vocabulary extracted from the Universal Dependencies project[6] and from Wiktionary using the `wiktextract` library (Ylonen, 2022). For generating lexical rules, we randomly select 400 nouns, 300 adjectives, 20 verbs, 5 prepositions and 15 determiners from our vocabulary. These lexical items are then assigned to respective non-terminal symbols with probabilities following a Zipfian distribution. Since our sentences

---

[6]We sample the words in our PCFGs from the dataset found at https://github.com/UniversalDependencies/UD_French-GSD.

are tokenised into words, we ensure that the lexical rules to generate `lm_dev`, `probe_train` and `probe_test` only contain words that were present in `lm_train`. This approach ensures that we avoid incorporating out-of-vocabulary words in the sets used for inference.

## A.1 PCFG for the first experiment

In this section, we report the PCFGs used in the experiment "Decoupling contextual gender information from static gender associations." We categorise feminine and masculine nouns into four groups based on their appearance in gendered (G) or gender-ambiguous (A) contexts within `lm_train` and whether they are present in a gendered context (G) in `probe_train` or remain unseen (U) in this set. The results presented in Table 1 concern exclusively nouns in the latter category. Thus, any gender information detected by the linguistic probe at inference time is encoded in the LM's representation and not merely memorised by the probe.

The following PCFG is used to generate `lm_train`.

```
S -> NP VP "."[1.0]

PP -> PREP NP [1.0]
VP -> VERB [0.5] | VERB NP [0.5]

NP -> NPGend [0.4] | NPAmb [0.4]
NP -> NP PP [0.2]

NPGend -> DETFem npGendFem [0.5]
NPGend -> DETMasc npGendMasc [0.5]
npGendFem -> NOUNFemGend [0.4]
npGendFem -> ADJFem NOUNFemGend [0.3]
npGendFem -> NOUNFemGend ADJFem [0.3]
npGendMasc -> NOUNMascGend [0.4]
npGendMasc -> ADJMasc NOUNMascGend [0.3]
npGendMasc -> NOUNMascGend ADJMasc [0.3]

NPAmb -> DETEpic npAmbFem [0.5]
NPAmb -> DETEpic npAmbMasc [0.5]
npAmbFem -> NOUNFemAmb [0.4]
npAmbFem -> ADJEpic NOUNFemAmb [0.3]
npAmbFem -> NOUNFemAmb ADJEpic [0.3]
npAmbMasc -> NOUNMascAmb [0.4]
npAmbMasc -> ADJEpic NOUNMascAmb [0.3]
npAmbMasc -> NOUNMascAmb ADJEpic [0.3]

NOUNFemAmb -> NOUNFemAG [0.5]
NOUNFemAmb -> NOUNFemAU [0.5]
```

```
NOUNMascAmb -> NOUNMascAG [0.5]
NOUNMascAmb -> NOUNMascAU [0.5]
NOUNFemGend -> NOUNFemGG [0.5]
NOUNFemGend -> NOUNFemGU [0.5]
NOUNMascGend -> NOUNMascGG [0.5]
NOUNMascGend -> NOUNMascGU [0.5]
```

The PCFG presented below is employed for generating `probe_train`.

```
S -> NP VP "."[1.0]

PP -> PREP NP [1.0]
VP -> VERB [0.5] | VERB NP [0.5]

NP -> NPGend [0.8] | NP PP [0.20]

NPGend -> DETFem npGendFem [0.5]
NPGend -> DETMasc npGendMasc [0.5]
npGendFem -> NOUNFemGend [0.4]
npGendFem -> ADJFem NOUNFemGend [0.3]
npGendFem -> NOUNFemGend ADJFem [0.3]
npGendMasc -> NOUNMascGend [0.4]
npGendMasc -> ADJMasc NOUNMascGend [0.3]
npGendMasc -> NOUNMascGend ADJMasc [0.3]

NOUNFemGend -> NOUNFemGG [0.5]
NOUNFemGend -> NOUNFemAG [0.5]
NOUNMascGend -> NOUNMascGG [0.5]
NOUNMascGend -> NOUNMascAG [0.5]
```

To create `probe_test`, we use variations of the following rules. Placeholder `X` denotes the context in which a noun appears in `lm_train` and can assume the values `G` or `A`. On the other hand, `Y` represents the context in `probe_train` and can be either `G` or `U`.

```
S -> NP VP "."[1.0]

PP -> PREP NP [1.0]
VP -> VERB [0.5] | VERB NP [0.5]

NOUNFem -> NOUNFemXY [1.0]
NOUNMasc -> NOUNMascXY [1.0]
```

If the selected noun category is to be tested in gendered contexts during inference, the following rules are also added to the PCFG.

```
NP -> NPGend [0.8] | NP PP [0.20]

NPGend -> DETFem npGendFem [0.5]
NPGend -> DETMasc npGendMasc [0.5]
npGendFem -> NOUNFem [0.4]
```

```
npGendFem -> ADJFem NOUNFem [0.3]
npGendFem -> NOUNFem ADJFem [0.3]
npGendMasc -> NOUNMasc [0.4]
npGendMasc -> ADJMasc NOUNMasc [0.3]
npGendMasc -> NOUNMasc ADJMasc [0.3]
```

If the nouns are to appear in gender-ambiguous contexts, on the other hand, the rules that follow are applied.

```
NP -> NPAmb [0.8] | NP PP [0.20]

NPAmb -> DETEpic npAmbFem [0.5]
NPAmb -> DETEpic npAmbMasc [0.5]
npAmbFem -> NOUNFem [0.4]
npAmbFem -> ADJEpic NOUNFem [0.3]
npAmbFem -> NOUNFem ADJEpic [0.3]
npAmbMasc -> NOUNMasc [0.4]
npAmbMasc -> ADJEpic NOUNMasc [0.3]
npAmbMasc -> NOUNMasc ADJEpic [0.3]
```

## A.2  PCFG for the second experiment

In this section, we report the PCFGs used in the experiment "Assignment of gender to epicene nouns."

The following PCFG is used to generate `lm_train`.

```
S -> NP VP "."[1.0]

PP -> PREP NP [1.0]
VP -> VERB [0.5] | VERB NP [0.5]

NP -> NPGend [0.4] | NPAmb [0.4]
NP -> NP PP [0.20]

NPAmb -> DETEpic NOUN [0.4]
NPAmb -> DETEpic ADJEpic NOUN [0.3]
NPAmb -> DETEpic NOUN ADJEpic [0.3]
NOUN -> NOUNMasc [0.35] | NOUNFem [0.35]
NOUN -> NOUN25 [0.1] | NOUN50 [0.1]
NOUN -> NOUN75 [0.1]

NPGend -> NPFem [0.35] | NPMasc [0.35]
NPGend -> NP25 [0.1] | NP50 [0.1]
NPGend -> NP75 [0.1]

NPFem -> DETFem NOUNFem [0.4]
NPFem -> DETFem ADJFem NOUNFem [0.3]
NPFem -> DETFem NOUNFem ADJFem [0.3]
NPMasc -> DETMasc NOUNMasc [0.4]
NPMasc -> DETMasc ADJMasc NOUNMasc [0.3]
NPMasc -> DETMasc NOUNMasc ADJMasc [0.3]
```

```
NP25 -> DETFem np25Fem [0.25]          np25Fem -> NOUN25 ADJFem [0.4]
NP25 -> DETMasc np25Masc [0.75]        np25Masc -> NOUN25 [0.4]
np25Fem -> NOUN25 [0.4]                np25Masc -> ADJMasc NOUN25 [0.3]
np25Fem -> ADJFem NOUN25 [0.3]         np25Masc -> NOUN25 ADJMasc [0.3]
np25Fem -> NOUN25 ADJFem [0.3]
np25Masc -> NOUN25 [0.4]               NP50 -> DETFem np50Fem [0.50]
np25Masc -> ADJMasc NOUN25 [0.3]       NP50 -> DETMasc np50Masc [0.50]
np25Masc -> NOUN25 ADJMasc [0.3]       np50Fem -> NOUN50 [0.4]
                                       np50Fem -> ADJFem NOUN50 [0.3]
NP50 -> DETFem np50Fem [0.50]          np50Fem -> NOUN50 ADJFem [0.3]
NP50 -> DETMasc np50Masc [0.50]        np50Masc -> NOUN50 [0.4]
np50Fem -> NOUN50 [0.4]                np50Masc -> ADJMasc NOUN50 [0.3]
np50Fem -> ADJFem NOUN50 [0.3]         np50Masc -> NOUN50 ADJMasc [0.3]
np50Fem -> NOUN50 ADJFem [0.3]
np50Masc -> NOUN50 [0.4]
np50Masc -> ADJMasc NOUN50 [0.3]       NP75 -> DETFem np75Fem [0.75]
np50Masc -> NOUN50 ADJMasc [0.3]       NP75 -> DETMasc np75Masc [0.25]
                                       np75Fem -> NOUN75 [0.4]
NP75 -> DETFem np75Fem [0.75]          np75Fem -> ADJFem NOUN75 [0.3]
NP75 -> DETMasc np75Masc [0.25]        np75Fem -> NOUN75 ADJFem [0.3]
np75Fem -> NOUN75 [0.4]                np75Masc -> NOUN75 [0.4]
np75Fem -> ADJFem NOUN75 [0.3]         np75Masc -> ADJMasc NOUN75 [0.3]
np75Fem -> NOUN75 ADJFem [0.3]         np75Masc -> NOUN75 ADJMasc [0.3]
np75Masc -> NOUN75 [0.4]
np75Masc -> ADJMasc NOUN75 [0.3]
np75Masc -> NOUN75 ADJMasc [0.3]
```

The subsequent PCFG is used for generating `probe_test`. The placeholders Y and X can be substituted with specific values to represent the noun category being tested and the context in which these nouns should appear, respectively. For Y, the options include Fem, Masc, 25, 50, or 75. As for X, it can be replaced with Fem, Masc, or Amb.

The PCFG described next is used in the generation of `probe_train`.

```
S -> NP VP "."[1.0]

PP -> PREP NP [1.0]
VP -> VERB [0.5] | VERB NP [0.5]

NP -> NPGend [0.8] | NP PP [0.20]

NPGend -> NPFem [0.35] | NPMasc [0.35]
NPGend -> NP25 [0.1] | NP50 [0.1]
NPGend -> NP75 [0.1]

NPFem -> DETFem NOUNFem [0.4]
NPFem -> DETFem ADJFem NOUNFem [0.3]
NPFem -> DETFem NOUNFem ADJFem [0.3]
NPMasc -> DETMasc NOUNMasc [0.4]
NPMasc -> DETMasc ADJMasc NOUNMasc [0.3]
NPMasc -> DETMasc NOUNMasc ADJMasc [0.3]

NP25 -> DETFem np25Fem [0.25]
NP25 -> DETMasc np25Masc [0.75]
np25Fem -> NOUN25 [0.4]
np25Fem -> ADJFem NOUN25 [0.3]
```

```
S -> NP VP "."[1.0]

PP -> PREP NP [1.0]
VP -> VERB [0.5] | VERB NP [0.5]

NP -> NPX [0.80] | NP PP [0.20]

NPAmb -> DETEpic NOUN [0.4]
NPAmb -> DETEpic ADJEpic NOUN [0.3]
NPAmb -> DETEpic NOUN ADJEpic [0.3]
NPFem -> DETFem NOUN [0.4]
NPFem -> DETFem ADJFem NOUN [0.3]
NPFem -> DETFem NOUN [0.4]
NPMasc -> DETMasc NOUN [0.4]
NPMasc -> DETMasc ADJMasc NOUN [0.3]
NPMasc -> DETMasc NOUN ADJMasc [0.3]

NOUN -> NOUNY [1.0]
```

## B  Transformer Language Model

We used the same model size as in (White and Cotterell, 2021), with an embedding size and an output size of 256, 3 hidden layers with 4 attention heads each and feed-forward networks with a hidden size of 1024. We also experimented with larger models, such as the original transformer size in (Vaswani et al., 2017), but we found that it did not significantly improve perplexity or the probe's accuracy. Therefore, we opted for smaller models to save computational resources and training time. Our models were trained for 100 epochs, with one warmup epoch. We used a learning rate of 0.0005, a dropout rate of 0.3 and a batch size of 64. The weights from the epoch that performed best on the development set were selected as the final model.

To ensure the statistical significance of our findings and mitigate the influence of model variance, we conducted each experiment by training 20 distinct language models and probes. The resulting probe accuracies were averaged, allowing us to compute 95% confidence intervals for our measurements.