# OpenReview forum: "Using Artificial French Data to Understand the Emergence of Gender Bias in Transformer Language Models"
_EMNLP/2023/Conference — EMNLP 2023 Main_

### Official Review · Reviewer_B4CF · 2023-07-21

**Soundness:** 4

**Excitement:**

4: Strong: This paper deepens the understanding of some phenomenon or lowers the barriers to an existing research direction.

**Paper Topic And Main Contributions:**

This work focuses on the problem of how neural language models (NLMs) discover information. In particular, this work takes an initial step toward exploring the less researched topic of how neural models discover linguistic properties of words, such as gender, as well as the rules governing their usage. By evaluating the NLM's capacities on a test set specifically designed to contain examples that differ from those of the training set based on well-defined criteria, this work finds that NLMs can memorize gender information from the training data, they also learn to use sentence structure to infer the gender of a word occurrence.

Overall, this work studies an interesting problem concerning the emergence of gender bias in Transformer language models. The findings challenge the assumption that training data imbalance is the sole cause of gender bias in NLMs. This work may shed light on the salient factors influencing the emergence of gender bias in NLMs.

**Reasons To Accept:**

1. The problem studied in this work is critical and the analysis is in good scientific rigor with convincing experimental results and insightful analysis. The paper is written in a clear and concise style.

2. This work provides a novel perspective on how gender bias emerges in language models. The work may help determine under which conditions a model correctly captures gender information or, on the contrary, appears gender-biased.

**Reasons To Reject:**

This is overall a solid short paper. I do not find any major concerns.

**Reproducibility:**

4: Could mostly reproduce the results, but there may be some variation because of sample variance or minor variations in their interpretation of the protocol or method.

**Reviewer Confidence:**

4: Quite sure. I tried to check the important points carefully. It's unlikely, though conceivable, that I missed something that should affect my ratings.

---

> ### Author Rebuttal · Authors · 2023-08-27
>
> Dear Reviewer B4CF,
>
> Thank you for your positive feedback! We are pleased that you found our work novel, rigorous and convincing.

---

### Official Review · Reviewer_WdXg · 2023-07-28

**Soundness:** 3

**Excitement:**

2: Mediocre: This paper makes marginal contributions (vs non-contemporaneous work), so I would rather not see it in the conference.

**Missing References:**

The paper would benefit from being better rooted within bias research, for example:

- *How Does Grammatical Gender Affect Noun Representations in Gender-Marking Languages?* - Gonen et al. (2019)
- *Language (Technology) is Power: A Critical Survey of “Bias” in NLP* - Blodgett et al. (2020)
- *The Birth of Bias: A case study on the evolution of gender bias in an English language model* - Van der Wal et al. (2022)

For probing there is a extensive line of work that has raised doubts about interpreting probing accuracy as a direct measure for model performance. Useful references here could be:

- *Amnesic Probing: Behavioral Explanation with Amnesic Counterfactuals* - Elazar et al. (2020)
- *Designing and Interpreting Probes with Control Tasks* - Hewitt et al. (2019)

**Paper Topic And Main Contributions:**

The paper presents an investigation into gender bias using artificial languages. It is often said that one of the major causes for gender bias in LMs stems from unbalance in the training data, and the paper proposes to investigate this in a controlled setup using PCFGs.

It is shown that contrary to previous work, gender unbalances are not amplified. Furthermore, it is shown that gender bias can persist even in the case where the training data is completely balanced.

**Questions For The Authors:**

- Why was it not possible to extract the masculine/feminine noun constructions from natural data? I expect that the simplicity of these constructions would make this possible, and it would remove the caveat of the constructions being semantically implausible.

**Reasons To Accept:**

- A focused and systematic approach to investigate model bias in isolation.
- Surprising finding that biases do not seem to be amplified as has been reported in earlier work on bias.
- The usage of artificial languages for model interpretability is a fruitful line of work, and I am excited to seeing it be applied in a gender bias context.

**Reasons To Reject:**

- The paper barely engages with the current literature on (gender) bias.
- The paper presents little to no pointers how its findings can be of use to analysis of bias in LLMs in general.
- The paper does not engage with the literature on probing, which has shown that solely using accuracy as a metric is insufficient and not necessarily reflective of actual model behaviour.
- The context-free assumption of the PCFGs loses the semantic plausibility of the constructions, which might make it challenging to relate these findings to natural settings. A formal language setting that takes this better into account would strengthen the results.

**Reproducibility:**

3: Could reproduce the results with some difficulty. The settings of parameters are underspecified or subjectively determined; the training/evaluation data are not widely available.

**Reviewer Confidence:**

4: Quite sure. I tried to check the important points carefully. It's unlikely, though conceivable, that I missed something that should affect my ratings.

**Typos Grammar Style And Presentation Improvements:**

Table 2 could perhaps be better represented as a line plot, right now the trend is much harder to interpret.

---

> ### Author Rebuttal · Authors · 2023-08-27
>
> Dear Reviewer WdXg,
>
> Thank you for your detailed review and for the reference pointers you provided. We address each of your concerns separately below.
>
> > “The paper does not engage with the literature on probing, which has shown that solely using accuracy as a metric is insufficient and not necessarily reflective of actual model behaviour.”
>
> We are well aware of the limitations of using probe accuracy highlighted in the papers you put forward and have discussed the problems raised by the use of probes in our “Limitations” section. Hewitt et al. (2019) call attention to the fact that by relying solely on probe accuracy we cannot tell whether a high score means the information is encoded in the representations or whether the probe has just learnt the task. The strategy we have used to avoid this problem is explained in footnote 4: “to ensure that a high probe accuracy indicates the encoding of gender information within the language model’s representations rather than its memorisation by the probe, nouns present in probe_train are excluded from probe_test.”
>
> Elazar et al. (2020) highlight that behavioural conclusions should not be drawn from probing results. We agree that our results do not make it possible to draw conclusions about the behaviour of language models with regard to gender. However, the aim of our work is to shed light on how transformer LMs are able to learn gender and, in the process, identify the possible causes of bias in this family of models, rather than to determine exactly why a particular model behaves in a biased way. We believe probes to be a fitting tool to study this question.
>
> >“The context-free assumption of the PCFGs loses the semantic plausibility of the constructions, which might make it challenging to relate these findings to natural settings. A formal language setting that takes this better into account would strengthen the results.”
>
> > “Why was it not possible to extract the masculine/feminine noun constructions from natural data? I expect that the simplicity of these constructions would make this possible, and it would remove the caveat of the constructions being semantically implausible.”
>
> We explain our choice to use PCFGs in Appendix A (l. 470):
> “In our case, we chose to disregard the semantics of our artificial languages, focusing solely on the grammatical aspects of gender. This deliberate choice serves to narrow down the range of possibilities for how bias can manifest and emerge, at least for the present.”
> Thanks to the additional page in the final version, we will be able to move this discussion to the “main” paper as, as you pointed out, it is an important question. Gulordava et al. (2018) show that syntax can be learnt independently from semantics, so we chose to focus only on the grammatical aspect of gender since semantics could introduce potential confounding factors (such as stereotypes introduced by the distributional environment of words) and we wanted to isolate the factors we were testing, such as the grammatical informativity of the context at training and inference time and the overall gender distribution in the training data. This is also why we did not extract our constructions from natural data, but generated them using PCFGs instead.
>
> > “The paper barely engages with the current literature on (gender) bias.”
>
> We are familiar with the references you provided but we did not elaborate on these points due to lack of space. We will use the extra page in the final version to address these issues more thoroughly.
> We have, for example, taken into account Blodgett et al. (2020)’s suggestion that papers on bias explicitly state what is meant by the word bias and their normative assumptions about what model behaviours are good and which ones are bad. In the section “Influence of the gender distribution on default gender guessing” (l. 246), we define precisely what we mean by “degree of bias” and what would be the expected results for a biased and an unbiased model. We do, however, agree that it would be beneficial to further expand on these topics making our assumptions about why these model behaviours are bad and who exactly would be the wronged party explicit instead of taking them to be self-evident.
>
> > “The paper presents little to no pointers how its findings can be of use to analysis of bias in LLMs in general.”
>
> Please see the response to Reviewer bPtj for this point.
>
> Lastly, thank you for the suggestion to turn Table 2 into a line plot, it does indeed make it a lot clearer and easier to interpret. We will use this representation in our final version.

---

### Official Review · Reviewer_bPtj · 2023-08-04

**Soundness:** 4

**Excitement:**

4: Strong: This paper deepens the understanding of some phenomenon or lowers the barriers to an existing research direction.

**Paper Topic And Main Contributions:**

By using an artificial, controlled dataset, this work studies how well Transformer language models are able to capture gender information under different circumstances (which are controlled creating artificial data using PCFG). Despite that this has only studied using French and using artificial data (thus, results cannot be extrapolated to other languages and/or data), the most interesting conclusion is that "the origin
of bias in language models is more complex than initially thought".

**Reasons To Accept:**

* Interesting study that could benefit future studies on the field.

**Reasons To Reject:**

* Results cannot be directly extrapolated.

**Reproducibility:**

3: Could reproduce the results with some difficulty. The settings of parameters are underspecified or subjectively determined; the training/evaluation data are not widely available.

**Reviewer Confidence:**

4: Quite sure. I tried to check the important points carefully. It's unlikely, though conceivable, that I missed something that should affect my ratings.

**Typos Grammar Style And Presentation Improvements:**

l. 019: (Manning et al., 2020; Belinkov et al., 2020) -> (Belinkov et al., 2020; Manning et al., 2020)
l. 042: a NLM -> an NLM
l. 050: methodology of -> methodology from/by
l. 070: (White and Cotterell, 2021; Kim and Linzen, 2020) -> (Kim and Linzen, 2020; White and Cotterell, 2021)
l. 126: LM of -> LM from/by
l. 197: of -> by

---

> ### Author Rebuttal · Authors · 2023-08-27
>
> Dear Reviewer bPtj,
>
> Thank you for your valuable time, feedback and style and grammar corrections.
>
> > “this has only studied using French and using artificial data (thus, results cannot be extrapolated to other languages and/or data)”
>
> The goal of this short paper is to investigate whether some precise factors can influence a transformer-based LM’s ability to assign gender by working in a controlled setting, and **not** to make claims about why real-world language models trained on natural data are biased, which would be impossible using artificial data. We believe that this first objective is a necessary step towards understanding the biases in the models used today and that by making it possible to carry out experiments under perfectly controlled conditions, the methodology we are introducing allows us to draw general conclusions about the capacities of the transformer architecture. For example, our results show that having an unbalanced gender distribution in the training data does not necessarily lead a transformer-based LM to over-assign the majority class gender. This suggests that transformers’ learning process is not inherently biased or irremediably predisposed to reproducing the most frequent patterns.
> We agree that further work is necessary to assess if our conclusions still hold for models trained on natural language or for languages with different gender systems and these are points we discussed in the “Limitations” section. However, we think the results we present here are already important and can help better direct these future works, by suggesting factors whose influence could be tested and a methodology for doing so.
>
> Concerning reproducibility, our code and dataset will be made available upon publication, which should make it easy to reproduce our results. We will also add the average runtime and number of parameters in the camera-ready version.

---

### Meta-Review · Area_Chair_Ytzq · 2023-09-08

**Recommendation:** 5

**Metareview:**

This paper introduces an artificially developed French corpus to better understand the circumstances under which transformer models capture gender information. Its findings are especially impactful, suggesting that gender biases can persist even when training corpora appear fully balanced. This contradicts previous research which often attributes gender biases to imbalanced training data. While the paper was acknowledged as sound (two 4s, one 3), there are potential issues in regards to the extensibility of this work to other languages and the lack of engagement with existing research on gender bias. The authors addressed these concerns in their rebuttals, however, and acknowledged that missing discussions would be included in the final version of the paper.

In light of this, only minor revisions, addressing the aforementioned concerns, need to be made to ensure this paper is camera ready.

*As an additional note for the authors, I highly recommend adding to the title and at minimum, the abstract that the corpus was made for French. I would also like to reiterate Reviewer WdXg's recommendation to explicitly state how you believe "experiments on artificial data will ultimately lead to 'understanding the biases in the models used today'."

---

### Decision · Program_Chairs · 2023-10-07

**Decision:**

Accept-Main

**Comment:**

This paper introduces an artificially developed French corpus to better understand the circumstances under which transformer models capture gender information. Its findings are especially impactful, suggesting that gender biases can persist even when training corpora appear fully balanced. This contradicts previous research which often attributes gender biases to imbalanced training data. While the paper was acknowledged as sound (two 4s, one 3), there are potential issues in regards to the extensibility of this work to other languages and the lack of engagement with existing research on gender bias. The authors addressed these concerns in their rebuttals, however, and acknowledged that missing discussions would be included in the final version of the paper.

In light of this, only minor revisions, addressing the aforementioned concerns, need to be made to ensure this paper is camera ready.

*As an additional note for the authors, I highly recommend adding to the title and at minimum, the abstract that the corpus was made for French. I would also like to reiterate Reviewer WdXg's recommendation to explicitly state how you believe "experiments on artificial data will ultimately lead to 'understanding the biases in the models used today'."